# Design of a Unified Algorithm to Ensure the Sustainable Use of Air Transport during a Pandemic

**Stanislav Szabo** 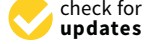, **Sebastián Makó** *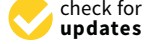, **Michaela Kešeľová** 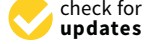 and **Stanislav Szabo, Jr.**

Department of Air Traffic Management, Faculty of Aeronautics, Technical University of Košice,
040 01 Košice, Slovakia; stanislav.szabo@tuke.sk (S.S.); michaela.keselova@tuke.sk (M.K.);
stanislav.szabo.2@tuke.sk (S.S.J.)
* Correspondence: sebastian.mako@tuke.sk

**Abstract:** The COVID-19 pandemic has had a significant impact on air transport in various parts of the world. The impact of the pandemic has been and still is significant within the Member States of the European Union. The introduction focused on identifying and monitoring the pandemic spread in the individual Member States. The research focused on two periods that were compared with each other based on key indicators, i.e., reproduction rate, hospitalized patients, or ICU patients. Identification and monitoring of the above-mentioned periods were performed by an observational study of collected data mentioned below. Subsequently, an algorithm was proposed, which was to determine an index number of a given country based on key indicators mentioned earlier. The index number is an assessment of the pandemic situation in a given country. The index number calculation in the monitored periods divided the countries into two groups: countries with the index number higher than one and countries with the index number lower than one. The latter can continue using air transport by pandemic situation assessment conducted by the algorithm. The air transport utilization rate depends on the second part of the algorithm, where the allowed number of routes is calculated for individual airlines. The use of an algorithm for calculating the index number of individual countries and at the same time monitoring the development of key indicators every 14 days is a suitable method for ensuring the sustainable use of air transport to minimize financial losses.

**Keywords:** algorithm; pandemic; air transport; routes

## 1. Introduction

The coronavirus disease 2019 (COVID-19) is a transmissible and infectious disease caused by severe acute respiratory syndrome coronavirus 2 (SARS-CoV-2). The disease was first diagnosed in December 2019 in Wuhan, the capital of China's Hubei province, and has expanded globally, resulting in the ongoing coronavirus pandemic [1,2]. The virus can be spread between humans by contact with biological fluids (for instance, through coughing and sneezing), mainly through close contact from person to person and between people who are physically near each other. Thus, the disease is highly contagious [1]. In February 2020, data from the World Health Organization (WHO) had confirmed that more than 43,000 cases had been recognised in 28 countries/regions, with >99% of patients being identified in China [3]. The WHO declared COVID-19 as the sixth public health emergency of international concern. SARS-CoV-2 is closely related to two bat-derived severe acute respiratory syndrome-like coronaviruses, bat-SL-CoVZC45 and bat-SL-CoVZXC21. The infection has been estimated to have a mean incubation period of 6.4 days and a primary reproduction rate of 2.24–3.58. Among patients with pneumonia caused by SARS-CoV-2, fever was the most common symptom, followed by cough [4,5]. Originating from China, COVID cases quickly spread worldwide, urging world governments to implement stringent measures to isolate patients and limit the transmission rate of the virus [6]. All nations and countries have implemented various strategies to handle emerging issues [7].

Governments in the European Union have used travel bans, the closure of borders, and limitations on people's mobility to decrease the spread of the virus [7,8]. Intense government measures, including travel restrictions and travel bans, have been implemented to limit the spread of COVID-19 worldwide. Throughout the early stages of the pandemic, global mobility modulated the initial outbreak pattern. Global mobility, including tourists, business people, sportspeople, and many others, increases the chances of a virus outbreak in different places in a very short time due to worldwide connections between various destinations that can be traveled to within hours. Multiple studies have shown a close relationship between mobility and the spreading of contagious disease, particularly during the early stages of an outbreak [9,10]. Many authors [11–14] claim that travel restrictions are beneficial in the early or initial stage of an outbreak when confined to a certain area that is a major source of the spread. However, travel restrictions may be less efficient once the outbreak is more widespread [15]. With a limited medical capacity to treat the disease, non-pharmaceutical interventions (NPI) are the primary strategy to contain the pandemic. Unprecedented global travel restrictions and "stay-at-home" orders are causing the most significant disruption of the global economy. With international travel bans affecting over 90% of the world population and widespread restrictions on public gatherings and community mobility, tourism largely ceased in March 2020 [16]. Most airlines decided to operate a regular schedule until drastic mobility restrictions stopped them. This caused unexpected drops in flight numbers from mid-March 2020, when lockdowns, border closures, and travel bans began to be the principal policy response across Europe and America [17,18]. According to IATA [19], it is possible to categorize flights into three groups: high-risk flights, medium-risk flights, and low-risk flights. Prevention and control measures are implemented after a thorough and complete evaluation of the outbreak at the flights' area of origin [20]. Amongst the restrictive measures and rules, international air traffic and flight suspension are undoubtedly effective in reducing mobility globally in the short term. Still, it also has a major long-term and short-term socioeconomic impact [21]. The aviation industry and airports face a considerable decline in revenues due to the lower number of flights [22,23]. Airliners, especially in the European Union, were also forced to cancel charter flights. Tourism which is directly connected to Airliners and their performance indicators is also one of the central aspects of achieving economic growth and one of the world's largest and fastest-growing economic activity industries. Many developing countries also see tourism as a large part of economic growth and sustainability policies and as a source of limited financial services, employment, foreign exchange gains, and technological assistance [24,25]. Beyond this, in an era of enormous change, reflecting the outcomes of the COVID-19 pandemic, the essence of sustainable transport in the continual development of tourism is of critical importance. Air transport, as the large propagator of global tourism via fast, safe, long-distance travel, has, of course, changed significantly [26]. Research conducted shows that air transport is directly connected to state policy in terms of restrictions during COVID-19. However, no key or pattern is currently known as a universal solution to ensure sustainable use of air transport during the pandemic.

## 2. Materials and Methods

The COVID-19 pandemic has affected the aviation industry worldwide. The influence of the COVID-19 pandemic persists within the European Union, mainly because the Member States of the European Union have not applied the same restrictions. Restrictions were different in each Member State, which was caused by the unequal spread of the pandemic in the Member States. Overall, restrictions were applied to travel, sport, education, and leisure activities to prevent the spread of the pandemic by restricting contact between large groups of people.

### 2.1. Monitoring the Spread of the Pandemic in the Member States of the European Union

At the beginning of the research, it was important to collect the necessary data on the development of the pandemic in each country [27,28]. These data show the spread of the

pandemic during selected periods in chronological order [29,30]. The particular periods point to the speed and diversity of spread of the pandemic in the individual Member States of the European Union. Not all Member States of the European Union specify the exact number of beds with oxygen support and ICU units. For research purposes and based on the data mentioned above, the number of beds with oxygen support was estimated to be 1/10 of the original number of all hospital beds in the Member States of the European Union. The number of ICU units was estimated to be 15 thousandths of the actual number of all hospital beds in the Member States of the European Union. It should be noted that the spread of the COVID-19 pandemic also affected the number of beds with oxygen support and ICU units.

The basic reproduction rate was used to measure the transmission potential of a disease. It was the average number of secondary infections produced by a typical case of an infection in a population where everyone is susceptible [31]. In Europe, the second wave of the pandemic started to get serious at the end of September and the beginning of October (Figure 1), when there was a significant increase in the reproduction rate.

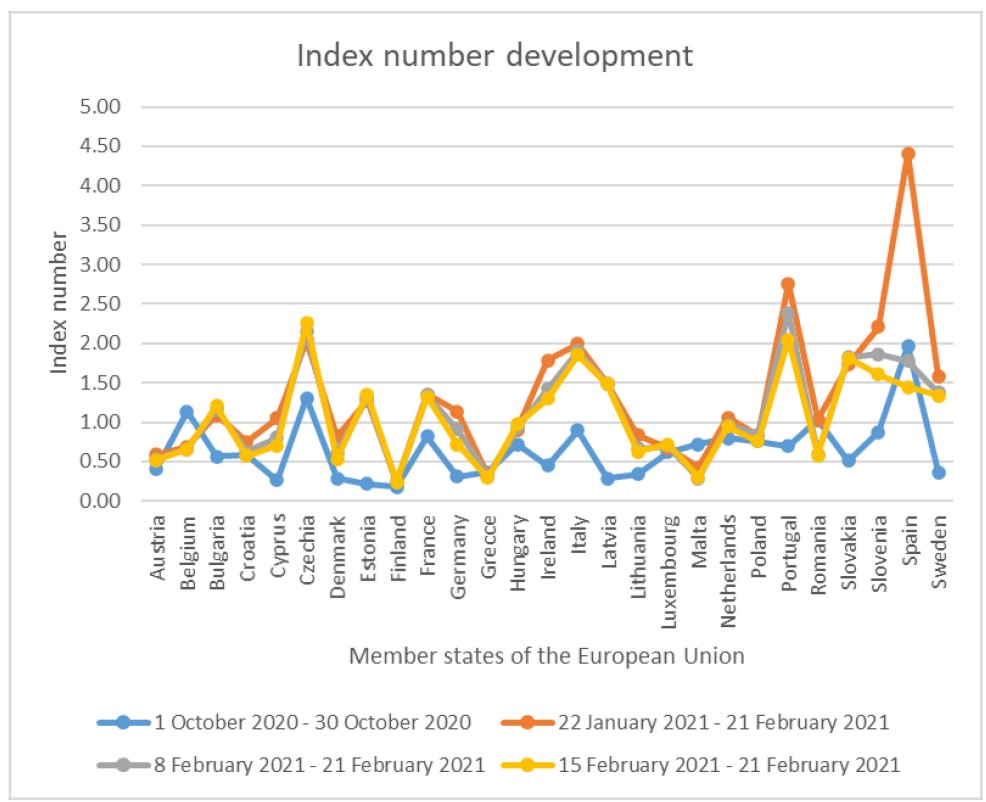

**Figure 1.** Index number development.

As was stated in (Table 1), the numbers of hospitalized patients or ICU patients were not high enough to force the Member States of the European Union to stop air traffic completely. At that time, individual states began to take steps to limit the spread of the pandemic. Spain and the Czech Republic were higher among the surveyed countries. The pandemic spread during October 2020 was chosen as a reference sample when comparing other pandemic spread data in January and February. The values given in all tables were the average values of individual days of the observed period.

In further monitoring of the pandemic, research was focused on monitoring changes in key indicators of the pandemic spread in January and February. From 22 January 2021 to 21 February 2021 (Table 2), a significant increase in hospitalized patients and ICU patients were seen compared to data from October 2020. The reproduction rate had a decreasing tendency. After applying the algorithm and considering all the variables, it was clear that most countries should not have been connected by air transport during this period (Figure 1).

**Table 1.** Key indicators for monitoring the spread of a pandemic in the period from 1 October 2020 to 30 October 2020 [27–30].

| Country | New Cases | Reprod. Number | Hospit. Patients | ICU Patients | Hospital Beds with Oxygen Support * | ICU Beds * | People Vaccinated | Population |
|---|---|---|---|---|---|---|---|---|
| Austria | 1939 | 1.36 | 726 | 142 | 6637 | 996 | N/A | 9,046,000 |
| Belgium | 10,025 | 1.43 | 2793 | 473 | 6537 | 981 | N/A | 11,589,616 |
| Bulgaria | 1033 | 1.50 | 1478 | 89 | 5179 | 777 | N/A | 6,948,445 |
| Croatia | 1056 | 1.52 | 562 | 56 | 2274 | 341 | N/A | 4,105,268 |
| Cyprus | 84 | 1.51 | 18 | 2 | 298 | 45 | N/A | 875,899 |
| Czechia | 8527 | 1.38 | 3768 | 592 | 7100 | 1065 | N/A | 10,708,982 |
| Denmark | 593 | 1.16 | 119 | 17 | 1448 | 218 | N/A | 5,792,203 |
| Estonia | 49 | 1.17 | 35 | 4 | 622 | 93 | N/A | 1,326,539 |
| Finland | 197 | 1.20 | 51 | 8 | 1817 | 273 | N/A | 5,540,718 |
| France | 26,080 | 1.26 | 11,939 | 2010 | 39,033 | 5855 | N/A | 65,273,512 |
| Germany | 7706 | 0.38 | 12,013 | 863 | 67,027 | 10,055 | N/A | 83,783,945 |
| Greece | 670 | 1.27 | 530 | 70 | 4388 | 659 | N/A | 10,423,056 |
| Hungary | 1576 | 1.24 | 1764 | 325 | 6782 | 1017 | N/A | 9,660,350 |
| Ireland | 816 | 1.20 | 245 | 32 | 1461 | 219 | N/A | 4,937,796 |
| Italy | 11,760 | 1.57 | 8629 | 785 | 19,227 | 2885 | N/A | 60,461,828 |
| Latvia | 131 | 1.42 | 87 | 8 | 1051 | 158 | N/A | 1,886,202 |
| Lithuania | 327 | 1.43 | 254 | 11 | 1786 | 269 | N/A | 2,722,291 |
| Luxembourg | 278 | 1.46 | 67 | 8 | 282 | 42 | N/A | 625,976 |
| Malta | 96 | 1.24 | 47 | 10 | 198 | 30 | N/A | 441,539 |
| Netherlands | 7473 | 1.27 | 1233 | 349 | 5689 | 854 | N/A | 17,134,873 |
| Poland | 8749 | 1.56 | 7843 | 980 | 25,054 | 3758 | N/A | 37,846,605 |
| Portugal | 2121 | 1.34 | 1144 | 166 | 3457 | 792 | N/A | 10,196,707 |
| Romania | 3670 | 1.26 | 6872 | 716 | 13,259 | 1988 | N/A | 19,237,682 |
| Slovakia | 1533 | 1.36 | 581 | 76 | 3176 | 477 | N/A | 5,459,643 |
| Slovenia | 923 | 1.52 | 300 | 50 | 936 | 141 | N/A | 2,078,932 |
| Spain | 13,435 | 1.18 | 12,958 | 1848 | 13,886 | 2084 | N/A | 46,754,783 |
| Sweden | 1016 | 1.33 | 274 | 32 | 2242 | 336 | N/A | 10,099,270 |

* Estimated value.

**Table 2.** Key indicators for monitoring the spread of a pandemic in the period from 22 January 2021 to 21 February 2021 [27–30].

| Country | New Cases | Reprod. Number | Hospit. Patients | ICU Patients | Hospital Beds with Oxygen Support * | ICU Beds* | People Vaccinated | Population |
|---|---|---|---|---|---|---|---|---|
| Austria | 1470 | 0.98 | 1294 | 287 | 6637 | 996 | 121,216 | 9,046,000 |
| Belgium | 2182 | 1.03 | 1728 | 312 | 6537 | 981 | 232,861 | 11,589,616 |
| Bulgaria | 736 | 1.14 | 3141 | 283 | 5179 | 777 | 21,496 | 6,948,445 |
| Croatia | 407 | 0.85 | 1148 | 56 | 2274 | 341 | 64,951 | 4,105,268 |
| Cyprus | 117 | 0.86 | 117 | 26 | 298 | 45 | 15,322 | 875,899 |
| Czechia | 7606 | 1.03 | 6032 | 1098 | 7100 | 1065 | 189,566 | 10,708,982 |
| Denmark | 494 | 0.83 | 490 | 91 | 1448 | 218 | 180,625 | 5,792,203 |
| Estonia | 594 | 1.10 | 465 | 36 | 622 | 93 | 23,523 | 1,326,539 |
| Finland | 393 | 1.10 | 130 | 21 | 1817 | 273 | 91,260 | 5,540,718 |
| France | 19,946 | 1.01 | 26,841 | 3223 | 39,033 | 5855 | 959,716 | 65,273,512 |
| Germany | 9214 | 0.88 | 49,050 | 3249 | 67,027 | 10,055 | 1,526,605 | 83,783,945 |
| Greece | 946 | 1.16 | 539 | 54 | 4388 | 659 | 141,298 | 10,423,056 |
| Hungary | 1528 | 1.10 | 3800 | 240 | 6782 | 1017 | 143,184 | 9,660,350 |
| Ireland | 1069 | 0.73 | 1272 | 186 | 1461 | 219 | 143,000 | 4,937,796 |
| Italy | 12,291 | 0.98 | 21,855 | 2183 | 19,227 | 2885 | 1,326,263 | 60,461,828 |
| Latvia | 734 | 1.00 | 1100 | 49 | 1051 | 158 | 16,779 | 1,886,202 |
| Lithuania | 650 | 0.82 | 1166 | 26 | 1786 | 269 | 59,170 | 2,722,291 |
| Luxembourg | 144 | 1.08 | 69 | 13 | 282 | 42 | 7309 | 625,976 |
| Malta | 149 | 1.00 | 26 | 6 | 198 | 30 | 17,767 | 441,539 |
| Netherlands | 4041 | 0.91 | 1506 | 586 | 5689 | 854 | 135,000 | 17,134,873 |
| Poland | 5839 | 1.00 | 13,073 | 776 | 25,054 | 3758 | 638,798 | 37,846,605 |
| Portugal | 6528 | 0.72 | 5662 | 797 | 3457 | 792 | 210,734 | 10,196,707 |
| Romania | 2449 | 0.97 | 5945 | 975 | 13,259 | 1988 | 374,681 | 19,237,682 |
| Slovakia | 1965 | 1.01 | 3422 | 253 | 3176 | 477 | 99,455 | 5,459,643 |
| Slovenia | 983 | 0.90 | 917 | 157 | 936 | 141 | 52,340 | 2,078,932 |
| Spain | 21,821 | 0.94 | 28,967 | 4599 | 13,886 | 2084 | 1,097,369 | 46,754,783 |
| Sweden | 2846 | 1.21 | 1613 | 246 | 2242 | 336 | 221,504 | 10,099,270 |

* Estimated value.

The reproduction rate should not be the only and determining factor. A comprehensive assessment of the situation was needed. Spain has had the worst results in research, and a nationwide lockdown, including air traffic, was to be applied in the country to limit the spread of the disease. The 30-day average of key indicators was a long period for monitoring changes in the spread of the pandemic, and, therefore, in the next section, a 14-day average and a 7-day average were presented.

Subsequently, the research was focused on the period from 8 February 2021 to 21 February 2021 (Table 3). While conducting the research and utilizing the algorithm, it was agreed that the 14-day average of the pandemic spread key indicators were more crucial for ensuring sustainable air transport. When comparing the values of Tables 2 and 3, relatively minor changes could be noticed.

**Table 3.** Key indicators for monitoring the spread of a pandemic in the period from 8 February 2021 to 21 February 2021 [27–30].

| Country | New Cases | Reprod. Number | Hospit. Patients | ICU Patients | Hospital Beds with Oxygen Support * | ICU Beds * | People Vaccinated | Population |
|---|---|---|---|---|---|---|---|---|
| Austria | 1538 | 1.07 | 1121 | 264 | 6637 | 996 | 212,062 | 9,046,000 |
| Belgium | 2062 | 1.02 | 1621 | 309 | 6537 | 981 | 349,992 | 11,589,616 |
| Bulgaria | 923 | 1.22 | 3416 | 292 | 5179 | 777 | 41,407 | 6,948,445 |
| Croatia | 325 | 0.89 | 941 | 43 | 2274 | 341 | 61,241 | 4,105,268 |
| Cyprus | 107 | 0.99 | 87 | 19 | 298 | 45 | 22,813 | 875,899 |
| Czechia | 8442 | 1.12 | 6168 | 1188 | 7100 | 1065 | 266,445 | 10,708,982 |
| Denmark | 423 | 0.90 | 341 | 67 | 1448 | 218 | 199,439 | 5,792,203 |
| Estonia | 692 | 1.17 | 495 | 35 | 622 | 93 | 35,917 | 1,326,539 |
| Finland | 438 | 1.11 | 125 | 21 | 1817 | 273 | 170,641 | 5,540,718 |
| France | 19,194 | 0.97 | 26,453 | 3342 | 39,033 | 5855 | 1,922,706 | 65,273,512 |
| Germany | 7346 | 0.89 | 30,401 | 3936 | 67,027 | 10,055 | 2,380,609 | 83,783,945 |
| Greece | 1133 | 1.17 | 539 | 71 | 4388 | 659 | 332,812 | 10,423,056 |
| Hungary | 1895 | 1.25 | 3908 | 242 | 6782 | 1017 | 291,396 | 9,660,350 |
| Ireland | 821 | 0.79 | 906 | 162 | 1461 | 219 | 154,900 | 4,937,796 |
| Italy | 12,322 | 1.03 | 20,626 | 2089 | 19,227 | 2885 | 1,464,945 | 60,461,828 |
| Latvia | 694 | 1.00 | 955 | 71 | 1051 | 158 | 18,075 | 1,886,202 |
| Lithuania | 520 | 0.87 | 901 | 23 | 1786 | 269 | 79,268 | 2,722,291 |
| Luxembourg | 162 | 1.10 | 71 | 14 | 282 | 42 | 14,089 | 625,976 |
| Malta | 158 | 1.07 | 16 | 4 | 198 | 30 | 30,235 | 441,539 |
| Netherlands | 3679 | 0.98 | 1395 | 531 | 5689 | 854 | 414,858 | 17,134,873 |
| Poland | 6322 | 1.12 | 12,329 | 956 | 25,054 | 3758 | 1,291,569 | 37,846,605 |
| Portugal | 2294 | 0.51 | 4724 | 771 | 3457 | 792 | 291,758 | 10,196,707 |
| Romania | 2456 | 1.03 | 4999 | 267 | 13,259 | 1988 | 661,062 | 19,237,682 |
| Slovakia | 2058 | 1.03 | 3576 | 281 | 3176 | 477 | 193,218 | 5,459,643 |
| Slovenia | 790 | 0.90 | 737 | 137 | 936 | 141 | 55,684 | 2,078,932 |
| Spain | 13,652 | 0.81 | 16,236 | 1094 | 13,886 | 2084 | 1,328,459 | 46,754,783 |
| Sweden | 3079 | 1.21 | 1337 | 222 | 2242 | 336 | 326,607 | 10,099,270 |

* Estimated value.

However, after applying the algorithm these changes were significant enough to limit some countries using air transport. As was shown, it was possible to ensure air transport sustainability within the restrictions in countries with low key indicators.

The last period was focused on the period from 15 February 2021 to 21 February 2021 (Table 4). Changes between the 14-day average and the 7-day average were minimal. This fact was also reflected in the calculated values by applying the algorithm. For optimal operation of the algorithm, it was necessary to work with average values for the last 14 days. As was pointed out in Tables 2 and 3, other periods were too long or too short to actively monitor changes in the pandemic spread.

**Table 4.** Key indicators for monitoring the spread of a pandemic in the period from 15 February 2021 to 21 February 2021 [27–30].

| Country | New Cases | Reprod. Number | Hospit. Patients | ICU Patients | Hospital Beds with Oxygen Support * | ICU Beds * | People Vaccinated | Population |
|---|---|---|---|---|---|---|---|---|
| Austria | 1698 | 1.12 | 1048 | 261 | 6637 | 996 | 298,199 | 9,046,000 |
| Belgium | 2263 | 1.06 | 1604 | 318 | 6537 | 981 | 423,946 | 11,589,616 |
| Bulgaria | 998 | 1.23 | 3614 | 300 | 5179 | 777 | 92,381 | 6,948,445 |
| Croatia | 327 | 0.96 | 851 | 39 | 2274 | 341 | 87,169 | 4,105,268 |
| Cyprus | 107 | 1.05 | 81 | 16 | 298 | 45 | 37,570 | 875,899 |
| Czechia | 9307 | 1.17 | 6381 | 1273 | 7100 | 1065 | 341,687 | 10,708,982 |
| Denmark | 459 | 0.94 | 289 | 62 | 1448 | 218 | 320,891 | 5,792,203 |
| Estonia | 743 | 1.20 | 512 | 37 | 622 | 93 | 61,904 | 1,326,539 |
| Finland | 489 | 1.13 | 132 | 26 | 1817 | 273 | 287,998 | 5,540,718 |
| France | 20,056 | 0.96 | 25,794 | 3368 | 39,033 | 5855 | 2,564,530 | 65,273,512 |
| Germany | 7538 | 0.93 | 23,690 | 3029 | 67,027 | 10,055 | 3,335,830 | 83,783,945 |
| Greece | 1096 | 1.17 | 539 | 57 | 4388 | 659 | 467,656 | 10,423,056 |
| Hungary | 2223 | 1.33 | 4043 | 273 | 6782 | 1017 | 453,457 | 9,660,350 |
| Ireland | 782 | 0.82 | 799 | 153 | 1461 | 219 | 222,073 | 4,937,796 |
| Italy | 12,481 | 1.06 | 20,149 | 2067 | 19,227 | 2885 | 2,201,756 | 60,461,828 |
| Latvia | 686 | 1.00 | 828 | 91 | 1051 | 158 | 29,288 | 1,886,202 |
| Lithuania | 475 | 0.90 | 861 | 22 | 1786 | 269 | 122,410 | 2,722,291 |
| Luxembourg | 178 | 1.11 | 74 | 15 | 282 | 42 | 23,259 | 625,976 |
| Malta | 158 | 1.10 | 24 | 4 | 198 | 30 | 43,888 | 441,539 |
| Netherlands | 3955 | 1.04 | 1382 | 524 | 5689 | 854 | 778,744 | 17,134,873 |
| Poland | 7116 | 1.18 | 12,332 | 667 | 25,054 | 3758 | 1,785,194 | 37,846,605 |
| Portugal | 1681 | 0.49 | 3917 | 703 | 3457 | 792 | 429,020 | 10,196,707 |
| Romania | 2533 | 1.06 | 4831 | 266 | 13,259 | 1988 | 803,098 | 19,237,682 |
| Slovakia | 2066 | 1.05 | 3623 | 278 | 3176 | 477 | 276,535 | 5,459,643 |
| Slovenia | 755 | 0.91 | 638 | 119 | 936 | 141 | 95,070 | 2,078,932 |
| Spain | 11,012 | 0.78 | 13,020 | 925 | 13,886 | 2084 | 1,893,290 | 46,754,783 |
| Sweden | 3251 | 1.21 | 1288 | 217 | 2242 | 336 | 398,092 | 10,099,270 |

\* Estimated value.

### 2.2. The Effect of the Pandemic on Air Transport in the European Union

The impact of the pandemic on air transport was significant and persisted. The following subchapter presents the pandemic's impact on European airlines and airports by comparing 2020 and 2019 key performance indicators.

By comparing the key performance indicators of European airlines for the years 2020 and 2019, the authors were able to obtain a comprehensive view of the impact the pandemic had on individual companies. Results showed (Table 5) a decrease in sales, passengers, and load factor for all airlines. The only airline that did not report a loss in 2020 was the Hungarian Wizz Air, which increased its net profit by 79 mils. €. The total losses of the airlines amounted to tens of billions of euros. Airlines are currently dependent on government assistance and are compensating for financial losses by laying off staff.

Due to the decrease in the number of flights (Table 6) in 2020, airports recorded a decline in passengers' number compared to 2019. The reduction in the number of passengers was accentuated by regional policy decisions mentioned in Section 2.1, which resulted in the closure of the airports to reduce the risk of the spread of the pandemic. The average decrease in passengers in 2020 compared to 2019 was 74.24%. Airports without aircraft and passenger handling did not make a profit, which only created further financial losses in this sector.

**Table 5.** Comparison of key performance indicators of European airlines.

| Airliner | TR 2020 [1] | TR 2019 [1] | Net Income 2020 [1] | Net Income 2019 [1] | PAX 2020 [2] | PAX 2019 [2] | LF 2020 | LF 2019 |
|---|---|---|---|---|---|---|---|---|
| Ryanair Holding [32] | 340 | 1.91 | −306 | 88 | 25,200 | 121,600 | 72.0% | 96.0% |
| Lufthansa Group [33] | 13,589 | 36,424 | −6725 | 1213 | 36,354 | 145,299 | 63.2% | 82.6% |
| Air France-KML Group [34] | 11,088 | 27,189 | −7078 | 290 | 5211 | 87,624 | 41.4% | 87.9% |
| International Airlines Group [35] | 7806 | 25,51 | −6923 | 1715 | 31,275 | 118,253 | 63.8% | 84.6% |
| Austrian Airlines [35] | 460 | 2108 | −381 | 15 | 3114 | 14,613 | 61.9% | 80.8% |
| Brussels Airlines [35] | 414 | 1473 | −332 | −32 | 2362 | 10,285 | 68.3% | 81.5% |
| Wizz Air [36] | 2761 | 2319 | 344 | 265 | 40,027 | 34,566 | 93.6% | 92.8% |
| SAS Group [18] | 2009 | 4517 | −908 | 60 | 12,610 | 29,761 | 60.5% | 75.2% |
| Tap Portugal [37] | N/A | 3345 | N/A | −105 | N/A | 17,052 | N/A | 80.1% |
| Aegean Airlines [38] | N/A | 1308 | N/A | 78 | N/A | 14,900 | N/A | 85.0% |
| Finnair [39] | 829 | 3097 | −523 | 74 | 14,645 | 34,856 | 63.0% | 81.7% |
| TUI Group [40] | 7952 | 18,928 | −3139 | 532 | 8057 | 21,075 | N/A | N/A |
| Air Baltic [41] | 120 | 392 | −578 | −640 | 1179 | 3875 | 55.6% | 76.6% |
| Norwegian Air [42] | 897 | 4293 | −2272 | −158 | 6870 | 36,200 | 75.2% | 86.6% |

[1] in millions, [2] in thousands.

**Table 6.** Comparison of passenger statistics at major airports in Europe [43].

| State | IATA Code | Airport | PAX Statistics 2019 [1] | PAX Statistics 2020 [1] | Change |
|---|---|---|---|---|---|
| Austria | VIE | Vienna International Airport | 31.6 | 7.8 | −75.3% |
| Belgium | BRU | Brussels Airport | 26.3 | 6.7 | −74.0% |
| Bulgaria | SOF | Sofia Airport | 7.1 | 2.9 | −58.7% |
| Croatia | ZAG | Zagreb Airport | 3.3 | 0.9 | −72.0% |
| Cyprus | LCA | Larnaca Airport | 8.2 | 1.6 | −79.5% |
| Czechia | PRG | Václav Havel Airport Prague | 17.8 | 3.0 | −79.4% |
| Denmark | CPH | Copenhagen Airport | 30.1 | 7.5 | −75.0% |
| Estonia | TLL | Tallin Airport | 0.8 | 0.2 | −73.6% |
| Finland | HEL | Helsinki Airport | 26.0 | 5.0 | −76.8% |
| France | CDG | Paris Charles de Gaulle Airport | 76.2 | 33.1 | −69.4% |
| Germany | FRA | Frankfurt am Main Airport | 70.6 | 18.8 | −73.0% |
| Germany | MUC | Munich Airport | 47.9 | 11.1 | −76.8% |
| Greece | ATH | Athens International Airport | 25.6 | 8.1 | −68.4% |
| Hungary | BUD | Budapest Ferenc Liszt International Airport | 16.2 | 3.8 | −76.0% |
| Ireland | DUB | Dublin Airport | 32.9 | 7.4 | −78.0% |
| Italy | FCO | Rome–Fiumicino International Airport | 43.5 | 11.4 | −76.8% |
| Latvia | RIX | Riga International Airport | 7.8 | 0.5 | −91.0% |
| Lithuania | KUN | Kaunas International Airport | 1.2 | 0.4 | −68.0% |
| Luxembourg | LUX | Luxembourg Airport | 4.4 | 1.4 | −68.0% |
| Malta | MLA | Malta International Airport | 7.3 | 1.7 | −76.0% |
| Netherlands | AMS | Amsterdam Airport Schiphol | 71.0 | 20.8 | −71.0% |
| Poland | WAW | Warsaw Chopin Airport | 18.2 | 5.0 | −69.0% |
| Portugal | LIS | Lisbon Portela Airport | 31.0 | N/A | N/A |
| Romania | OTP | Henri Coandă International Airport | 14.7 | 4.4 | −69.0% |
| Slovakia | BTS | Bratislava Airport | 2.3 | 0.4 | −82.0% |
| Spain | MAD | Madrid-Barajas Airport | 61.7 | 17.1 | −79.7% |
| Sweden | ARN | Stockholm Arlanda Airport | 25.6 | 6.5 | −74.0% |

[1] in millions.

Figure 2 shows the estimated development of air traffic volume in the European Region for 2021 compared to 2020. The baseline values with which the values of 2020 and 2021 were compared were from 2019. The graph shows that Eurocontrol estimates a possible slight improvement in the Q2 of 2021. Air traffic percentages were low compared to 2019 and in 2021, and several organizations expected this trend to continue until individual states managed the fight against a pandemic. At the moment, the biggest concern was the speed of vaccination in the individual Member States of the European Union.

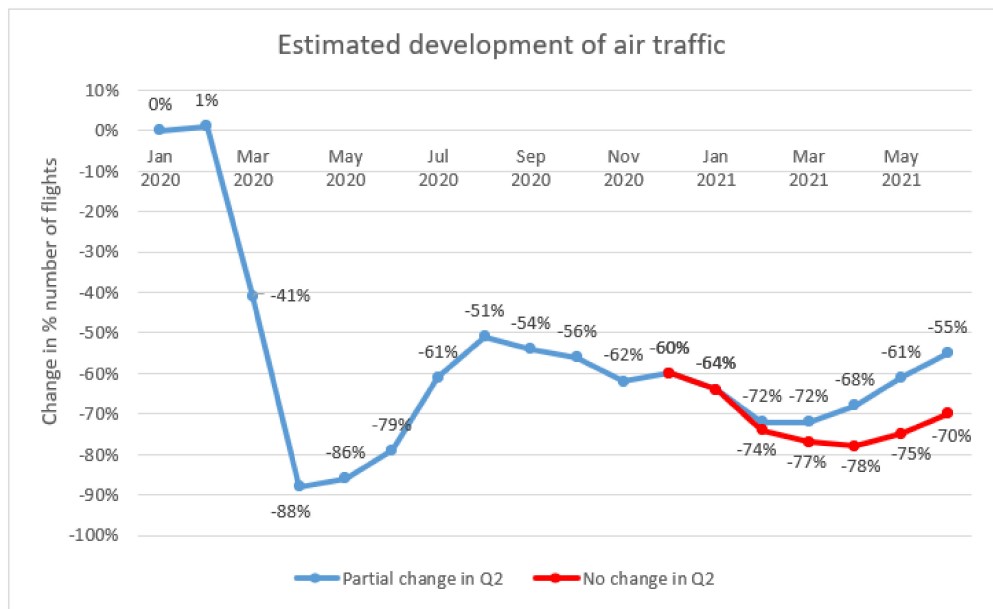

**Figure 2.** Estimated development of air traffic in Europe in Q2 of 2021.

This article was proposed to monitor changes in key performance indicators and design a unified algorithm, which was presented in Section 3, which would simplify airlines' operation by unifying the air transport rules. Unified rules for air transport within the Member States of the European Union would help to increase the number of passengers and speed up the recovery of air transport that experienced financial losses, as is shown in Tables 5 and 6.

## 3. Results

An algorithm (Figure 3) was proposed to unify the procedures for the use of air transport. The algorithm was focused on the evaluation of key indicators of individual countries and the calculation of the use of air transport within pandemic measures. The result of the algorithm was an index number of a particular country. The index number is an assessment of the pandemic situation in a particular country, and it determines whether the given country should operate air transport at all in a specific situation and, if so, to what extent.

Variables and constants: $H_{1i}$—hospital beds with oxygen support, $H_{2i}$—hospitalized patients, $H_{3i}$—ICU patients, $H_{4i}$—people vaccinated, $H_{5i}$—new cases, $H_{6i}$—ICU beds, $H_{7i}$—reproduction rate, $P_i$—country population, $P_k$—total number of countries, $P_{cm}$—number of passengers in 2019, $L_{cj}$—number of airline's routes within EU, L—total number of airlines, $H_i$—country index number, $A_i$—country index number shown at the end, $P_1$—counter of countries with index number higher than 1, $P_2$—counter of countries with index number lower than 1, $H_x$—country index counter with an index number lower than 1, x—counter of total number of countries, C—index number of all countries involved with index number lower than 1, $L_j$—total index number for airline, $L_x$—counter of airlines, $K_j$—maximum number of passengers, $P_{cj}$—average estimated number of passengers, $D_j$—maximum permitted number of passengers, $M_j$—maximum number of passengers in 2019, i—auxiliary variable, j—auxiliary variable, m—auxiliary variable.

The authors developed the algorithm based on the theory of maximization of the utilization of air transport during the pandemic to ensure a controlled and sustainable environment for customers and airliners.

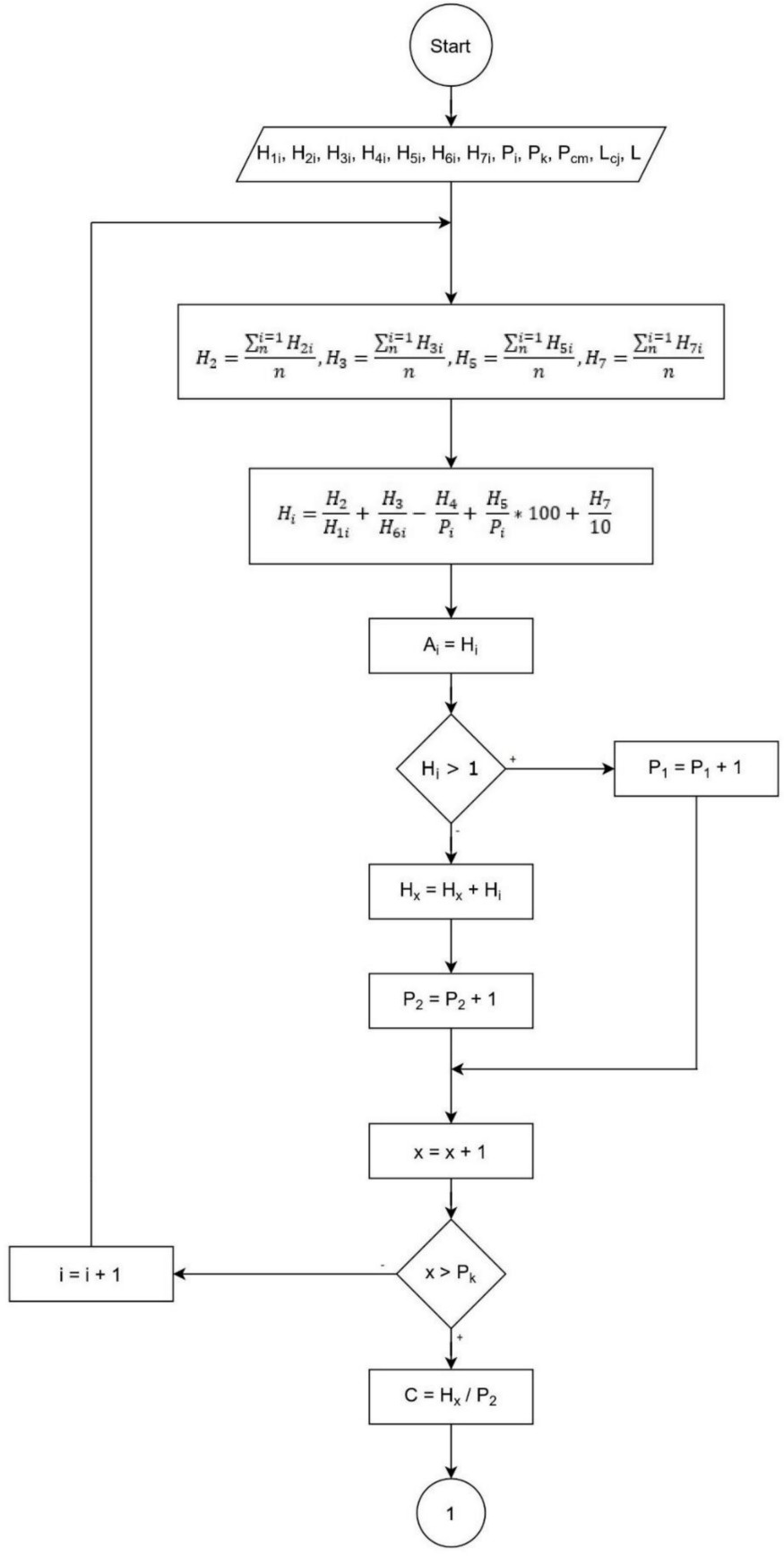

**Figure 3.** *Cont.*

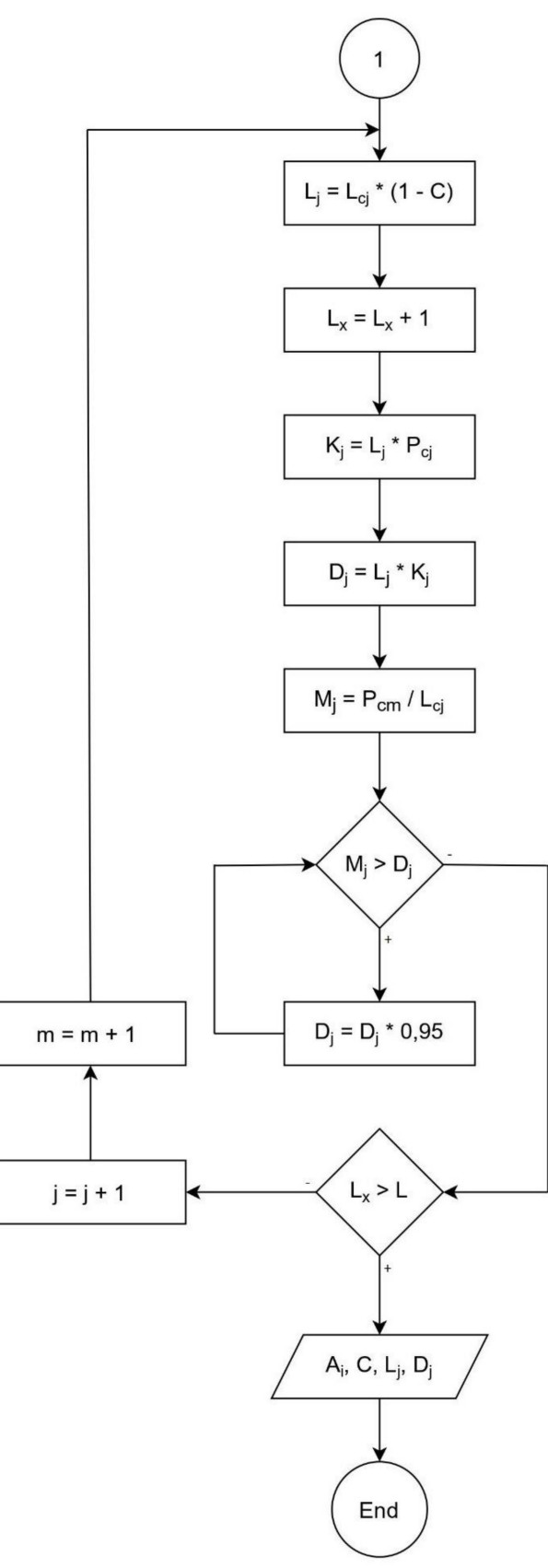

**Figure 3.** Unified algorithm for calculating usable capacities of air carriers with regard to the spread of a pandemic.

The main reason for creating the algorithm was to unify the procedure for assessing air transport possibilities during a pandemic. The European Union has not acted unanimously in this direction, which has caused heavy losses in the airline industry, especially in the European market. The proposed algorithm was divided into two parts. In the first part, based on the primary parameters H1–H7, the index number of a specific country was calculated. If the interval values were $0 < H_i < 1$, the country was included in the overall average of the assessed countries. If the value of $H_i > 1$, the country was not included in the assessed countries' overall average and was excluded from further calculations. Countries with an index number higher than 1 had a higher risk of transmitting the disease, which was no longer acceptable for air transport. The next step was the calculation of the C value, which was the ratio of countries with an index number lower than one to the number of these countries.

The second part of the algorithm focuses on calculating the air carrier's routes within the COVID-19 restrictions. The first important factor is how many routes, out of the original number of routes from 2019, can an air carrier use. To calculate this value, the original number of routes from 2019 was used along with the C value. This calculation was the value of $L_j$, which precisely defines the number of routes that the air carrier can use. A check calculation of the maximum number of passengers and the estimated number of passengers was presented in the algorithm to ensure compliance with hygienic and epidemiological measures. From the relationship $M_j > D_j$, the estimated number of passengers must not exceed the actual number of passengers from 2019, considering the number of lines.

The inconsistent approach to the COVID-19 restrictions by the individual Member States of the European Union continues to cause significant problems for air carriers. Other forms of transport have not been so significantly affected by the restrictions of individual countries. The algorithm considers all the necessary information to ensure the sustainable use of air transport, even considering the risks of a pandemic.

Testing and validation of the algorithm are based on data mentioned in Section 2.1 where all key indicators of the spread of the pandemic are mentioned as well as data from airliners about their lines and passengers in Member States of European Union.

By applying the algorithm for each period, the index numbers of individual countries within the European Union were calculated. To ensure sustainable air transport, all countries aim to have an index number value lower than 1. From the graph (Figure 1), it can be seen that in the first research period (Table 1) from 1 October 2020 to 30 October 2020, most countries had an index number value lower than 1, which creates an assumption for the sustainable use of air transport within the region. The reproduction rate was higher in this period than in the other periods examined, but the other key indicators examined were significantly lower.

Other research periods divided the countries into two groups. In the first group, there are countries with an index number higher than 1. Air transport should not be operated in these countries due to the higher risk of disease transmission. In the second group, some countries have an index number of less than 1. In these countries, part of the air transport operation as calculated in Figure 1 should be allowed. According to the proposed algorithm (Figure 3), countries such as Portugal, Spain, or the Czech Republic should not operate air transport, and the risk of spreading the disease is exceptionally high in these countries. Countries such as Austria, Finland, Greece, and Croatia should operate air transport with the constraints arising from Figure 1. By dividing the countries within the European Union, it is possible to ensure the sustainable use of air transport given the current state of pandemic spread in each country.

## 4. Discussion

The study points to the spread of a pandemic in the individual Member States of the European Union and at the same time suggests the use of a unified algorithm to ensure the sustainable use of air transport.

In the first phase, the research focused on specific pandemic spread periods in individual states. The reference point in the research was October when the pandemic was on the rise, but several indicators did not reach a critical value. Subsequently, they may compare these values with the values from January and February, thus the study focused on a 30-day, 14-day, and 7-day average. The research shows that the best explanatory value for applying the algorithm is exactly the 14-day average and the changes in it are relevant and suitable for the application of the calculation using the algorithm.

Due to the inconsistent approach of the Member States of the European Union to individual restrictions to reduce the spread of COVID-19, the unified algorithm was created. The design of the algorithm was based on pandemic spread evaluation in each Member State of the European Union conducted by monitoring of key indicators as well as monitoring the need for sustainable utilization of air transport. The algorithm was divided into two parts. The first part was focused on calculating the index number of individual countries. If the index number was lower than 1, the air traffic should be operated in the given country with the second part of the algorithm's restrictions. If the index number in a given country was higher than 1, air transport should not be operated in that country. The second part of the algorithm evaluates the possible usable capacity of air carriers concerning the index number. The evaluation of possible usable capacity was performed by calculation of index number and original capacity of the air carrier from 2019 in the specific country. The result of the evaluation process was certain percentage of possible usable capacity for the air carrier. Thus, the air carrier can provide air transport in the specific country to a certain level. At the same time, a check calculation was presented in the algorithm to ensure that the maximum number of passengers is never exceeded.

In the long run, the situation, especially in air transport, is critical. As is shown in previous chapters, the impact of the pandemic has caused many airlines billions of euros in losses. Research shows that the European Union could have approached all Member States uniformly, and the application of the same rules could have provided a better environment and sustainable utilization of air transport without restriction performed by individual governments.

Thus, it can be concluded that for the successful application of the algorithm in air transport, both airliners and the Member states of the European Union have to co-operate in terms of providing necessary information to ensure sustainable utilization of air transport during the pandemic. The application of the unified algorithm as proposed would help in this matter.

**Author Contributions:** Conceptualization, S.S., S.M., M.K. and S.S.J.; data curation, S.M. and M.K.; methodology, S.S. and S.S.J.; formal analysis, S.S., S.M., M.K. and S.S.J.; validation S.M. and S.S.J.; supervision, S.S. and S.M.; resources, M.K. and S.S.J.; writing—original draft preparation, S.M. and M.K.; writing—review and editing, S.S. and S.S.J. All authors have read and agreed to the published version of the manuscript.

**Funding:** This research received no external funding.

**Institutional Review Board Statement:** Not applicable.

**Informed Consent Statement:** Not applicable.

**Data Availability Statement:** Not applicable.

**Acknowledgments:** Not applicable.

**Conflicts of Interest:** The authors declare no conflict of interest.

## Abbreviations

| | |
|---|---|
| i.e., | id est |
| Reprod. | Reproduction |
| Hospit. | Hospitalized |
| ICU | Intensive care unit |
| TR | Total revenue |
| PAX | Passengers |
| LF | Load factor |
| IATA | International Air Transport Association |

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
