# Peer review of "Design of a Unified Algorithm to Ensure the Sustainable Use of Air Transport during a Pandemic"

_sustainability, doi:10.3390/su13115970_

Round 1

Reviewer 1 Report

The Covid-19 pandemic has had a significant impact on air transport in various parts of the world.  I recommend expanding the study beyond the Member States of the European Union. 

Author Response

Dear reviewer,

Thank you for your review. You mentioned that the Covid-19 pandemic has had a significant impact on air transport in various parts of the world and that you would recommend expanding the study beyond the Member States of the European Union.

We do agree with your statement that Covid-19 pandemic affected more countries than just countries in European union. However, adding more countries would immensely extend our paper. We also know that different parts of the world handled battle with the pandemic differently and thus the outcome of the research would be different.

Reviewer 2 Report

According to world statistics, Covid -19 has a negative impact on air traffic. It can be noticed that the number of flights, passengers and cargo is falling significantly, which has a direct impact on the condition of airlines. Therefore, the development of an algorithm to ensure sustainable air traffic is perfectly justified. Generally the paper is organized well. It contains all main parts of an original science paper. Chosen methods the studies  are suitable and efficiency. Nonetheless, I have got several remarks regarding the paper.

Below the most important ones are specified.

  1. In my opinion, the introduction is too long. contains a lot of obvious information, not necessarily related to the topic of the article. however, there is no information on similar solutions in the world. Authors should emphasize whether there is research on this subject or whether they have no knowledge.
  2. Table 1, 2, 3, 4 there is no definition of “reprod. Rate”
  3. Line 136 "a significant increase in hospitalized patients and ICU  patients were seen", there is no information what kind of test was used (ANOVA, 3 sigma)
  4. Table 1, 2, 3, 4, (ICU beds) there is no information what method of estimation was used
  5. Line 122., this sentence is incomprehensible.
  6. Line 154-157, a lack of reference data and what do “vital factors” mean
  7. Figure1, 3, no axes description

Author Response

Dear reviewer,

Thank you for your review. In next section we added, explained or clarified our statements according to your comments.

In my opinion, the introduction is too long. contains a lot of obvious information, not necessarily related to the topic of the article. however, there is no information on similar solutions in the world. Authors should emphasize whether there is research on this subject or whether they have no knowledge.

The introduction was shortened and reviewed to provide necessary information that supports the research. More information about the current state of the researched topic was added to the bottom part of introduction. Research conducted shows that air transport is directly connected to state policy in terms of restrictions during Covid-19. However, no key or pattern is currently known as a universal solution to ensure sustainable use of air transport during pandemic.

Deleted sections:

COVID-19 has globally affected populations, institutions and businesses, and organisations, involuntarily affecting the global economy's financial markets. Uncoordinated governmental measures and lockdowns have started a disruption in the supply chain [7] and have destroyed modern world economies' core sustaining pillars as global trade and cooperation succumbed to nationalist focus and competition for scarce supplies [6]. COVID-19 has affected many industries, such as Hospitality, tourism and aviation, the Real estate and housing sector, the Pharmaceutical industry, Information technology, media, research & development, the Food sector or the Sports industry [7].

COVID-19 pandemic has led to many industries coming to a standstill. It has led to restrictions of movement, many limitations and a travel ban. As a consequence of these restrictions, the transport sector, particularly in aviation, has badly impacted [22]. These interventions aim to decrease mobility and connections within the population and decrease the transmission of the virus, as measured by the adequate reproduction number (R, the average number of subsequent cases made by an original case). Many digital data sources showed human mobility reductions correlated well with decreases in COVID-19 incidence and social contacts at the beginning of the pandemic [23].

The main reason why the aviation industry is suffering the most, especially from the economic point of view, is the cancellation of domestic and international flights across the world to stop the spread of coronavirus [25].

The profit of airports has experienced severe contraction due to the imposition of travel restrictions. The global damage is already visible, and some airliners have already gone bankrupt. A significant number of countries worldwide have completely closed their borders for airline transportation. Passengers are either forbidden to travel or discouraged because of the countries' restrictions that they travel to that request quarantine [26]. Airlines are making every effort to stay in business as they present the essential task of connecting the world's economies.

Table 1, 2, 3, 4 there is no definition of “reprod. Rate”

Reprod. rate was changed to reprod. number and explanation was added under Table 1. The basic reproduction number (R0) is used to measure the transmission potential of a disease. It is the average number of secondary infections produced by a typical case of an infection in a population where everyone is susceptible.

Line 136 "a significant increase in hospitalized patients and ICU patients were seen", there is no information what kind of test was used (ANOVA, 3 sigma)

In revised manuscript sentence under Table 1. No test was performed. Sentence meaning is based on Table 1 and Table 2 data comparison. We also added this information to above-mentioned sentence. From 22 January 2021 to 21 February 2021 (Table 2), a significant increase in hospitalized patients and ICU patients was seen compare to data from October 2020.

Table 1, 2, 3, 4, (ICU beds) there is no information what method of estimation was used

Estimation method was added to chapter 2.1 above Table 1. Not all Member States of the European Union specify the exact number of beds with oxygen support and ICU units. For research purposes and based on the data mentioned above, the number of beds with oxygen support was estimated to be 1 tenth of the original number of all hospital beds in the Member States of European Union. The number of ICU units was estimated to be 15 thousandths of the actual number of all hospital beds in the Member States of the European Union. It should be noted that the spread of the Covid-19 pandemic also affected the number of beds with oxygen support and ICU units, which increased as needed.

Line 122., this sentence is incomprehensible.

Sentence was corrected. As was stated in (Figure 3) other indicators such as hospitalized patients or ICU patients were not high enough to force the Member States of the European Union to completely stop the air traffic.

Line 154-157, a lack of reference data and what do “vital factors” mean

References were included and the term “vital factors” was changed to key indicators. However, after applying the algorithm (Figure 2), these changes were significant enough to limit some countries using air transport (Figure 3). As is shown, it is possible to ensure air transport sustainability within the restrictions in countries with low key indicators.

Figure1, 3, no axes description

X and Y axis description was added to Figure 1 and Figure 3.

Reviewer 3 Report

Hi Authors,

The paper lacks rigor in terms of theoretical soundness and makes a lot of generalizations and redundant statements. The flow and logic of senteneces are not the best. Requires serious english language edits and the conclusions are not adeqautely supported by the analysis and application of algorithm. The authors will have to restructure the paper and provide theoretical framework and rationale that shows how an algorithm was developed and applied. I did not see significant findings that are novel and adds to extant literature.  It seemed more of a generic assumption of how variables related to COVID 19 adversely affected aviation in the EU and did not show any oustanding rationale for using the suggested algorithm. I  have more comments to the draft and attached a copy for guidance. I made suggestions for improvements. Thanks.

Author Response

Dear reviewer,

Round 2

Reviewer 3 Report

This version is a better improvement over the previous one. However, there are still some issues with grammar/spelling errors and sentence constructions. Some of the paragraphs have limted flow and makes it challenging to gauge the logic of the assertions. The acronyms used could have been introduced earlier in the draft to enable the reader get more acqauianted apriori. There are still challenges with how the algorithm was developed and how it is applied in practical sense. Further elucidation may be required. I recommend another round of proofing and edits to ensure soundness/coherence of the various paragraphs.

Author Response

Dear reviewer,

Thank you for your review. In next sections we added, explained or clarified our statements according to your comments.

You mentioned that there are still some issues with grammar/spelling errors and sentence constructions.

Introduction section – grammar mistakes corrected – lines 52 - 55: Originating from China, COVID cases quickly spread worldwide, urging world governments to implement stringent measures to isolate patients and limit the transmission rate of the virus.

Introduction section – grammar mistakes corrected – lines 59 - 60: Throughout the early stages of the pandemic, global mobility modulated the initial outbreak pattern.

Introduction section – grammar mistakes corrected – lines 65 - 67: Many authors [11], [12], [13], [14] claim that travel restrictions are beneficial in the early or initial stage of an outbreak when confined to a certain area that is a major source of the spread.

Introduction section – grammar mistakes corrected – lines 77 - 78: According to IATA [19], it is possible to categorise flights into three groups: high-risk flights, medium-risk flights, and low-risk flights.

Introduction section – grammar mistakes corrected – lines 82 - 83: Still, it also has a major long-term and short-term socioeconomic impact.

Introduction section – grammar mistakes corrected – lines 90 - 92: Beyond this, in an era of enormous change, reflecting the outcomes of the Covid-19 pandemic, the essence of sustainable transport in a continual development of tourism is of critical importance.

Materials and methods section – grammar mistakes corrected – lines 110 - 111: At the beginning of the research, it was important to collect the necessary data on development of the pandemic in each country [27,28].

Materials and methods section – grammar mistakes corrected – lines 178 - 179: As was pointed out in Table 2 and Table 4, other periods are too long or too short to actively monitor changes in the pandemic spread.

Materials and methods section – grammar mistakes corrected – lines 190 - 192: By comparing the key performance indicators of European airlines for the years 2020 and 2019, authors were able to obtain a comprehensive view of the impact pandemic has had on individual companies.

Results section – grammar mistakes corrected – lines 288 - 290: The reproduction rate was higher in this period than in the other periods examined, but the other key indicators examined were significantly lower.

Discussion section – grammar mistakes corrected – lines 309 - 310: The reference point in the research was October, when the pandemic was on the rise, but several indicators did not reach a critical value.

Discussion section – grammar mistakes corrected – lines 321 - 323: If the index number is lower than 1, the air traffic should be operated in the given country in accordance with the restrictions proposed in the second part of the algorithm.

You mentioned that some of the paragraphs have limited flow and makes it challenging to gauge the logic of the assertions.

Abstract section - statement corrected - lines 21 - 23:  The use of an algorithm for calculating the index number of individual countries and at the same time monitoring the development of key indicators every 14 days is a suitable method for ensuring the sustainable use of air transport to minimise financial losses.

Introduction section - statement corrected - lines 74 - 77: Most airlines decided to operate a regular schedule until drastic mobility restrictions stopped them. This caused unexpected drops in flight numbers from mid-March 2020, when lockdowns, border closures and travel bans began to be the principal policy response across Europe and America [17].

Materials and methods section - statement corrected - lines 119 - 121: It should be noted that the spread of the Covid-19 pandemic also affected the number of beds with oxygen support and ICU units.

Materials and methods section - statement corrected - lines 131 - 133: As was stated in (Figure 3) numbers of hospitalized patients or ICU patients were not high enough to force the Member States of the European Union to completely stop the air traffic

Materials and methods section - statement corrected - lines 144 - 145: In further monitoring of the pandemic, research was focused on monitoring changes in key indicators of the pandemic spread in January and February.

Materials and methods section - statement corrected - lines 153 - 155: The 30-day average of key indicators is a long period for monitoring changes in spread of the pandemic, and therefore, in the next section, a 14-day average and a 7-day average is presented.

Materials and methods section - statement corrected - lines 161 - 164: Subsequently, the research was focused on the period from 8 February 2021 to 21 February 2021 (Table 3). While conducting the research and utilizing the algorithm, it was agreed that the 14-day average of the pandemic spread key indicators is more crucial for ensuring sustainable air transport.

Materials and methods section - statement corrected - lines 202 - 204: The reduction in the number of the passengers was accentuated by regional policy decisions mentioned in section 2.1, which resulted in closure of the airports to reduce the risk of spread of the pandemic.

Materials and methods section - statement corrected - lines 212 - 213: The graph shows that Eurocontrol estimates a possible slight improvement in the Q2 of 2021.

Materials and methods section - statement corrected - lines 215 - 219: At the moment, the biggest concern is the speed of vaccination in the individual Member States of the European Union. This article was proposed to monitor changes in key performance indicators and design a unified algorithm, which was presented in Chapter 3, which would simplify airlines' operation by unifying the air transport rules.

Results section - statement corrected - lines 267 - 270: A check calculation of the maximum number of passengers and the estimated number of passengers was presented in the algorithm to ensure the compliance with hygienic and epidemiological measures.

Discussion section - statement corrected - lines 284 - 285: To ensure sustainable air transport, all countries aim to have an index number value lower than 1.

Discussion section - statement corrected - lines 323 - 324: If the index number in a given country is higher than 1, air transport should not be operated in that country.

You mentioned that the acronyms used could have been introduced earlier in the draft to enable the reader get more acqauianted apriori.

Position of the list of abbreviations was changed and the list is now below the abstract section.

You mentioned that there are still challenges with how the algorithm was developed and how it is applied in practical sense. Further elucidation may be required.

Discussion section – section added - lines 318 - 320: Design of the algorithm was based on pandemic spread evaluation in each Member State of the European Union done by monitoring of key indicators as well as monitoring the need of sustainable utilization of air transport.

Discussion section – section added - lines 326 - 331: The evaluation of possible usable capacity is done by calculation of index number and original capacity of the air carrier from 2019 in the specific country. The result of evaluation process is certain percentage of possible usable capacity for the air carrier. Thus, the air carrier can provide air transport in the specific country to a certain level. At the same time, a check calculation was presented in the algorithm to ensure that maximum number of passengers is never exceeded.